# The effect of HIV status on the frequency and severity of acute respiratory illness

**James Brown** [1,2]*, **Elisha Pickett**[2], **Colette Smith**[3], **Memory Sachikonye**[4], **Lucy Brooks**[5], **Tabitha Mahungu**[2], **David M. Lowe** [2,6], **Sara Madge**[2], **Mike Youle**[2], **Margaret Johnson**[2], **John R. Hurst**[1], **Timothy D. McHugh**[7], **Ibrahim Abubakar**[8], **Marc Lipman**[1,2]

**1** Division of Medicine, UCL Respiratory, University College London, London, England, United Kingdom, **2** Royal Free London NHS Foundation Trust, London, England, United Kingdom, **3** Centre for Clinical Research, Epidemiology, Modelling and Evaluation, Institute for Global Health, University College London, London, England, United Kingdom, **4** UK-CAB, HIV Treatment Advocates Network, London, England, United Kingdom, **5** The Keats Group Practice, London, England, United Kingdom, **6** Institute of Immunity and Transplantation, University College London, London, England, United Kingdom, **7** Division of Infection & Immunity, UCL Centre for Clinical Microbiology, University College London, London, England, United Kingdom, **8** Centre for Molecular Epidemiology and Translational Research, Institute for Global Health, University College London, London, England, United Kingdom

* james.brown13@nhs.net

## Abstract

### Introduction

Antiretroviral therapy has improved the health of people living with HIV (PLW-HIV), though less is known about how this impacts on acute respiratory illness. These illnesses are a common cause of ill health in the general population and any increase in their frequency or severity in PLW-HIV might have significant implications for health-related quality of life and the development of chronic respiratory disease.

### Methods

In a prospective observational cohort study following PLW-HIV and HIV negative participants for 12 months with weekly documentation of any acute respiratory illness, we compared the frequency, severity and healthcare use associated with acute respiratory illnesses to determine whether PLW-HIV continue to have a greater frequency or severity of such illnesses despite antiretroviral therapy.

### Results

We followed-up 136 HIV positive and 73 HIV negative participants for 12 months with weekly documentation of any new respiratory symptoms. We found that HIV status did not affect the frequency of acute respiratory illness: unadjusted incidence rates per person year of follow-up were 2.08 illnesses (95% CI 1.81–2.38) and 2.30 illnesses (1.94–2.70) in HIV positive and negative participants respectively, IRR 0.87 (0.70–1.07) p = 0.18. However, when acute respiratory illnesses occurred, PLW-HIV reported more severe symptoms (relative fold-change in symptom score 1.61 (1.28–2.02), p <0.001) and were more likely to seek healthcare advice (42% vs 18% of illnesses, odds ratio 3.32 (1.48–7.39), p = 0.003). After

**Data Availability Statement:** All relevant data are available at the University College London data repository (https://rdr.ucl.ac.uk/) with DOI 10.5522/04/11950284.v1.

**Funding:** The funding sources to be acknowledged are 1. National Institute for Health Research (Research Trainees Coordinating Centre) DRF-2015-08-210 2. British HIV Association (2015) The funders had no role in study design, data collection and analysis, decision to publish, or preparation of the manuscript.

**Competing interests:** The authors have declared that no competing interests exist.

adjustment for differences in baseline characteristics, PLW-HIV still had higher symptom scores when unwell.

## Conclusions

HIV suppression with antiretroviral therapy reduces the frequency of acute respiratory illness to background levels, however when these occur, they are associated with more severe self-reported symptoms and greater healthcare utilisation. Exploration of the reasons for this greater severity of acute respiratory illness may allow targeted interventions to improve the health of people living with HIV.

## Trial registration

ISRCTN registry (ISRCTN38386321).

## Introduction

The use of antiretroviral therapy (ART) has transformed the lives of people living with HIV (PLW-HIV). In populations with good access to ART and on-going care, PLW-HIV can now have a life expectancy equal to that of the general population.[1, 2]

Effective treatment is changing the demographics of the HIV positive population, who are increasing both in number and age.[3] There is a growing interest, therefore, in the extent to which they may be at greater risk of non-AIDS comorbidities such as chronic cardiovascular, neurological and respiratory diseases.[4–6] Studies evaluating respiratory health have reported a higher prevalence of chronic respiratory illness than the general population, [7] [8] which is only partly explained by greater exposure to established risk factors such as tobacco smoking. [9]

Although there is evidence that ART reduces the incidence of respiratory infections such as *Pneumocystis jirovecii* pneumonia, bacterial pneumonia and tuberculosis, [10, 11] most studies evaluating respiratory illness in the modern HIV population have been cross-sectional in nature and therefore cannot determine whether acute respiratory illnesses remain more frequent among PLW-HIV.[12, 13] PLW-HIV with Chronic Obstructive Pulmonary Disease (COPD) have a higher exacerbation frequency than equivalent HIV negative individuals.[14, 15] As acute respiratory illnesses are common in the general population and associated with significant morbidity and healthcare utilisation,[16] we postulated that if HIV positive individuals have a higher incidence of these illnesses (or if they were more severe or longer lasting) then this could impact significantly on health-related quality of life, productivity and healthcare costs. A greater frequency of acute respiratory illness might also contribute to the development of chronic lung disease in this population. If specific risk factors for acute respiratory illness could be identified, then interventions to reduce the incidence or severity of these illnesses (such as smoking cessation, immunisations and treatment for respiratory conditions) might be better targeted.

This study aimed to prospectively measure the frequency of acute respiratory illness among HIV positive individuals using antiretroviral therapy compared to HIV negative participants.

## Methods

### Study design, setting and population

We conducted a prospective cohort study of PLW-HIV and HIV negative individuals. We recruited participants between November 2015 and January 2017 from HIV care services and primary care in London, UK. This is a setting with a high uptake of antiretroviral therapy.[17]

We invited PLW-HIV to participate when attending ambulatory HIV-care appointments: the only eligibility criteria were age over 18 years, consent to participate and absence of symptoms of acute respiratory illness at study entry. We recruited HIV negative adults from primary care, using electronic medical records to invite potential participants with similar age, gender and smoking status to the expected characteristics of the HIV positive population sampled. To achieve this, primary care practice lists were used to identify all patients with age, gender and smoking status suitable for recruitment and individuals were randomly selected to be invited by post from these lists. There were no financial inducements offered, although we reimbursed limited travel expenses where appropriate. All participants provided written informed consent.

The study protocol was reviewed by the London Hampstead Research Ethics Committee (14/LO/1409) and registered with the ISRCTN registry (ISRCTN38386321).

### Procedures

At recruitment, participants completed a questionnaire detailing respiratory symptoms, prior illnesses, current treatment, tobacco and recreational drug use. We measured baseline respiratory health status using the St Georges Respiratory Questionnaire (SGRQ) and MRC dyspnoea scores.[18, 19]

Subjects underwent spirometry without bronchodilation, with standard ERS/ATS quality criteria,[20] using the Global Lung Function Initiative equations to calculate predicted values. [21] An obstructed airflow pattern was defined as an FEV1/FVC ratio <0.7, and restrictive pattern as FEV1/FVC >0.7 plus FVC <80% predicted.

We collected naso-pharyngeal swabs to assess the carriage of respiratory viral pathogens at baseline. HIV negative participants had their HIV status confirmed by blood test at recruitment.

### Detection of viral respiratory pathogens

The detection of respiratory viral pathogens from nasopharyngeal swabs at baseline and during an acute respiratory illness was undertaken using multiplex PCR testing. In-house real-time RT-PCR assay was performed which was designed to simultaneously detect: rhinovirus, influenza (A & B), parainfluenza (1–4), RSV, adenovirus, enterovirus, coronavirus (NL63, HKU, 229E, OC43), parechovirus and human metapneumovirus.

### Follow-up and identification of acute respiratory illnesses

We prospectively followed participants for 12 months and during this each participant was contacted weekly by email or telephone SMS message. In these weekly messages, participants were asked if they had any new respiratory symptoms indicative of an acute respiratory illness.

An acute respiratory illness was defined as the new occurrence (lasting more than 24 hours) of any of the following symptoms: cough, sore throat, blocked or runny nose with or without a sensation of facial pain or pressure, breathlessness or pain on breathing. This definition sought to include both upper and lower respiratory tract illnesses. No assumption was made about

whether an illness was caused by an infection, and fever was not included in the illness definition (although individuals were asked about a history of fever).

When participants reported new symptoms, their severity was assessed by supplementary questions including questions on daily activities and treatment. Symptoms were scored on a 0–6 scale and these scores were summarised by calculating a total score (with no weighting applied). We also invited participants to attend for review, (at which we collected samples for the detection of respiratory pathogens) and to complete a daily diary recording symptoms, medication usage and healthcare resource utilisation for the duration of their illness or up to 14 days, whichever was the longer. The questionnaires used on the webform and written bookets are provided as supplementary information.

## Sample size calculation

The primary study outcome was the number of acute respiratory tract illnesses occurring over the one-year follow up period. Based on data from the Pulmonary Complications of HIV study, Multicentre AIDS Cohort Study and the Women's Interagency HIV study we estimated that there would be at least a 50% higher than in HIV uninfected individuals without antiretroviral therapy,[22, 23] and planned the study size to detect lower incidence than this. To estimate the likely number of acute respiratory illnesses seen during follow up, we used data from the FluWatch study,[24] which reported that 44% of those without serological evidence of influenza infection experienced an acute respiratory illness each influenza season (the study was only conducted during influenza seasons). Based on this we conservatively estimated that at least 44% of the HIV negative participants will have an acute respiratory illness over a 12 month period. We planned to have a 2:1 ratio between HIV infected participants and controls. To have an 80% power to detect this difference (i.e. 68% vs. 45%) with a type 1 error of 5% we would need 119 HIV infected and 60 in the HIV uninfected participants in the cohort. We assumed up to a 20% drop out of subjects from the study then we plan to recruit 140 individuals with HIV infection and 70 individuals without HIV infection.

## Statistical analysis

The primary outcome was the difference in frequency of acute respiratory illness between HIV positive and negative participants. Pre-defined secondary outcomes were the severity and duration of acute respiratory illness, isolation of respiratory pathogens and healthcare utilisation.

We used Poisson regression models to analyse the frequency of acute respiratory illness (with weeks of data reported as an offset value). Continuous outcomes for symptom severity were analysed using linear regression models. We completed multivariable analysis, adjusting for potential confounding effects of factors chosen *a priori* based on known risk factors for acute respiratory illness. When participants did not answer the weekly study question, we assumed that they did not have an acute respiratory illness. We undertook data assembly and statistical analysis using Excel (Microsoft Ltd) and Stata v14, (Statacorp).

## Results

### Participants

209 individuals (136 HIV positive and 73 HIV negative) consented to participate in the study and submitted at least one week of follow-up data. Demographic details of study participants are given in Table 1. There were no significant differences in the age or gender of the HIV positive and negative participants, although there was a trend to suggest that PLW-HIV were

**Table 1. Participant baseline characteristics.**

| | | HIV Positive N = 136 | HIV Negative N = 73 | p value |
|---|---|---|---|---|
| **Gender** | Female, n (%) | 30 (22%) | 18 (25%) | 0.67[*] |
| | Male, n (%) | 106 (78%) | 55 (75%) | |
| **Age, years, mean (SD)** | | 50 (11) | 52 (8) | 0.11[**] |
| **Ethnicity** | Caucasian, n (%) | 103 (76%) | 70 (96%) | <0.01 [Δ] |
| | Black African / Caribbean, n (%) | 23 (17%) | 0 | |
| | South Asian, n (%) | 2 (1%) | 2 (3%) | |
| | Other, n (%) | 8 (6%) | 1 (1%) | |
| **Body mass index (BMI), kg/m$^2$ median (IQR)** | | 25 (23–28) | 25 (23–26) | 0.06 |
| **UK Born, n (%)** | | 73 (54%) | 54 (74%) | 0.07[*] |
| **Sexuality** | Heterosexual, n (%) | 37 (27%) | 57 (78%) | <0.001[Δ] |
| | Homosexual, n (%) | 86 (62%) | 16 (22%) | |
| | Bisexual, n (%) | 12 (9) | 0 | |
| | Not answered, n (%) | 1 (1%) | 0 | |
| **Educational attainment** | No qualifications, n (%) | 7 (5%) | 2 (3%) | 0.54[Δ] |
| | GCSE or equivalent qualifications at age 16, n (%) | 20 (15%) | 11 (15%) | |
| | A level or equivalent qualifications at age 18, n (%) | 24 (18%) | 8 (11%) | |
| | University education, n (%) | 72 (53%) | 44 (60%) | |
| | Other qualifications, n (%) | 13 (10%) | 7 (10%) | |
| | Not answered, n (%) | 0 | 1 (1%) | |
| **Immunisations (self-report)** | Influenza (last 12 months) | 90 (66%) | 21 (29%) | <0.01[*] |
| | Pneumococcal (ever) | 50 (37%) | 9 (12%) | <0.01[*] |
| **Comorbid conditions (self-report)** | Asthma | 22 (16%) | 7 (10%) | 0.21 [Δ] |
| | COPD | 3 (2%) | 1 (1%) | 0.56 [Δ] |
| | Heart disease | 5 (4%) | 2 (3%) | 1 [Δ] |
| **Previous history of respiratory opportunistic infection (HIV positive participants only)** | | 9 (7%) [##] | | |
| **Use of inhaled medications** | Any inhaled medication | 25 (18%) | 5 (7%) | 0.07 [Δ] |
| | Inhaled corticosteroids | 13 (10%) | 1 (1%) | 0.04 [Δ] |
| **Tobacco smoking** | Current smoker, n (%) | 39 (29%) | 12 (16%) | 0.08[*] |
| | Ex-smoker, n (%) | 46 (34%) | 34 (47%) | |
| | Never smoker, n (%) | 41 (37%) | 27 (37%) | |
| **Tobacco pack-years (median, IQR),[◇]** | | 6 (2–12) | 9 (3–15) | 0.17[#] |
| **Recreational drug use (ever), n (%)** | | 90 (66%) | 37 (51%) | <0.005[*] |
| **Recreational drug use (last 3 months), n (%)** | | 42 (31%) | 9 (12%) | 0.007[*] |
| **CD4 count, cells/μL, median (IQR)** | | 686 (458–848) | | |
| **Use of antiretroviral therapy during study** | | 136 (100%) | | |
| **HIV load < 40 copies/ml at baseline** | | 118 (87%) | | |

[*] Chi squared test,

[**] t—test

[#] Mann Whitney Test,

[Δ] Fisher's exact test

[◇] calculated for smokers and ex-smokers only

[##] consisting of *Pneumocystis jirovecii* pneumonia (7 cases), tuberculosis (1 case), non-tuberculous mycobacterial infection (1 case)

more likely to be current smokers (29% vs 16%, p = 0.08), and more likely to report previous or current (within the last 3 months) use of recreational drugs. All HIV positive participants used antiretroviral therapy during the study and 87% had an undetectable blood HIV load at baseline. The median (IQR) CD4 count was 686 (458–848) cells/μL.

**Table 2.  Baseline respiratory health status and spirometry.**

| | | HIV Positive (N = 136) | HIV Negative (N = 73) | p value |
|---|---|---|---|---|
| **Baseline St George's Respiratory Questionnaire score** | Symptoms, median (IQR) | 30 (8–45) | 11 (0–28) | <0.001[#] |
| | Activity, median (IQR) | 18 (6–36) | 6 (0–12) | <0.001[#] |
| | Impacts, median (IQR) | 4 (0–16) | 0 (0–2) | <0.001[#] |
| | Total score, median (IQR) | 13 (6–29) | 6 (2–9) | <0.001[#] |
| **Baseline MRC Dyspnoea score** | 1: Not troubled by breathlessness except on strenuous exercise, n (%) | 77 (57%) | 58 (81%) | 0.008[Δ] |
| | 2: Short of breath when hurrying on a level or when walking up a slight hill, n (%) | 43 (32%) | 12 (17%) | |
| | 3: Walks slower than most people on the level, stops after a mile or so, or stops after 15 minutes walking at own pace, n (%) | 6 (4%) | 0 | |
| | 4: Stops for breath after walking 100 yards, or after a few minutes on level ground, n (%) | 4 (3%) | 0 | |
| | 5: Too breathless to leave the house, or breathless when dressing/undressing, n (%) | 1 (1%) | 0 | |
| | Not answered | 3 (2%) | 2 (3%) | |
| **Spirometry***  | FEV1, L, mean (SD) | 3.22 (0.78) | 3.53 (0.73) | 0.01[□] |
| | FEV1% predicted, mean (SD) | 91% (14%) | 97% (11%) | 0.005[□] |
| | FVC, L, % predicted, mean (SD) | 4.16 (1.01) | 4.55 (0.98) | 0.02[□] |
| | FVC % predicted, mean (SD) | 93% (14%) | 99% (12%) | 0.02[□] |
| | Spirometry interpretation — Airflow obstruction, n %** | 13 (13%) | 5 (8%) | 0.004[Δ] |
| | Spirometry interpretation — Restriction, n% | 18 (19%) | 2 (3%) | |
| | Spirometry interpretation — Normal spirometry, n% | 65 (68%) | 54 (88%) | |

* 96 HIV positive and 61 HIV negative participants had spirometry results meeting ATS/ERS quality criteria

[#] Mann-Whitney test

[□] t-test

[Δ] Fisher's exact test

** FEV1/FVC <0.7

PLW-HIV reported worse respiratory health status at baseline with higher scores on the St George's Respiratory Questionnaire (median SGRQ Total score 13 (5–28) vs 6 (2–9), p< 0.001). Using the MRC dyspnoea scale, breathlessness also appeared to be more common: 59 (43%) vs 12 (17%) reporting MRC dyspnoea scale score of 2 or more (p <0.001). Spirometry was normal for most participants, but a greater proportion of PLW-HIV had abnormal spirometry at baseline (19% vs 3% with restrictive spirometry (FEV1/FVC ≥0.7 and FVC< 80% predicted)), and 13% vs 8% with obstructive spirometry (FEV1/FVC <0.7), p = 0.004 (Table 2).

## Detection of respiratory viral pathogens at baseline

Viral pathogens were identified in 5 (2%) of baseline nasopharyngeal swabs, all from PLW-HIV. These were: Parainfluenza 2 (two participants), Coronavirus OC43, Influenza A and Parechovirus. All other participants had negative baseline swab results.

## Follow-up and completion of weekly responses

The median number of weeks for which each participant provided data during follow-up was 44/52 weeks (85%); this was not significantly different between HIV positive and negative

participants. Sixty-six (42%) had 100% response rate to the weekly contacts and 60 (29%) had <80% response rate.

## Frequency of acute respiratory illness

One hundred and sixty-eight participants (80%) reported at least one acute respiratory illness during follow up, with the median number per person being 2 (range 0–7 illnesses).

There was no significant difference in the frequency of acute respiratory illness between HIV positive and negative participants. The unadjusted incidence rate per person year of follow-up was 2.08 (95% CI 1.81–2.38) in HIV positive and 2.30 (1.94–2.70) in HIV negative participants; IRR 0.87 (0.70–1.07 p = 0.18).

In univariable regression analyses, smoking status (being an ex-smoker), airflow obstruction (FEV1/FVC <0.7) and the presence of chronic respiratory symptoms (MRC score ≥2) at baseline were associated with a significantly greater frequency of acute respiratory illness. Participants of black ethnicity reported a lower frequency of events than white participants (IRR 0.64 (95% CI 0.42–0.96) p = 0.01) (Table 3).

In a multivariable model including HIV status, age, gender, ethnicity and the presence of spirometric abnormality, there was again no significant difference in the frequency of acute respiratory illness between HIV positive and negative participants (adjusted IRR 0.80 (0.61–1.06) p = 0.13).

## Severity and duration of illness

The web-based symptom questionnaire completed during an acute respiratory illness (by 90% of participants in 361 (97%) illnesses), demonstrated that PLW-HIV had a greater symptom severity score, with a median total score of 14 points (IQR 8–23) compared to 9 (5–14) in HIV negative participants (fold change in score 1.61 (1.28–2.02), p<0.001).

In addition, participants recorded written diaries detailing daily symptoms during 166 (45%) acute respiratory illnesses. In these, PLW-HIV also reported greater symptom scores with median total symptom scores per day of 9.36 points (IQR 5.77–14.95) versus 6.4 points (4.74–9.82) in HIV negative participants (p = 0.008). PLW-HIV also reported more days with at least mild symptoms, with a median duration of 8 (IQR 5.1–10.5) vs 6 (4.25–9.5) days—though this difference was not statistically significant (p = 0.18). The total symptom scores per day based on the written diaries provided by participants are displayed in Fig 1.

## Effect of participant baseline characteristics on symptom scores during acute respiratory illness

We explored the relationship between baseline characteristics and participant-reported severity of symptoms using the online responses. In univariable log-scale linear regression analyses, HIV status, airflow obstruction (FEV1/FVC <0.7), recent (within 3 months) use of recreational drugs and the presence of increased respiratory symptoms at baseline (MRC dyspnoea score or St George's Respiratory Questionnaire score) were associated with greater self-reported symptom severity during acute respiratory illness. HIV positive participants had a 61% greater symptom score in univariable analyses (unadjusted fold change in symptom score 1.61, 95% CI 1.28–2.02, p <0.001).

In a multivariable regression model assessing the direct effect of HIV status after adjustment for spirometric impairment, recent recreational drug use, SGRQ score and tobacco smoking, being HIV positive continued to be associated with a greater symptom score during acute respiratory illness, with an adjusted fold-change in symptom score of 1.50 (1.14–1.97, p = 0.004) (Table 4).

**Table 3. Risk factors for acute respiratory illness among cohort participants.**

| Characteristic | | N | Incidence rate of acute respiratory illness per person year of follow-up | Univariable analysis [b] IRR (95% CI) | P value | Multivariable analysis [b] IRR (95% CI) | P value |
|---|---|---|---|---|---|---|---|
| HIV status | HIV positive | 136 | 2.08 (1.81–2.38) | 0.87 (0.70–1.07) | 0.18 | 0.81 (0.61–1.06) | 0.13 |
| | HIV negative | 73 | 2.30 (1.94–2.70) | 1 | | 1 | |
| Gender | Female | 48 | 1.97 (1.56–2.48) | 0.84 (0.54–1.23) | 0.42 | 0.88 (0.65–1.21) | 0.45 |
| | Male | 161 | 2.21 (1.95–2.47) | 1 | | 1 | |
| Age (years) | 65+ | 24 | 1.98 (1.42–2.68) | 0.94 (0.67–1.32) | 0.40 | 0.87 (0.58–1.31) | 0.90 |
| | 55–65 | 35 | 2.19 (1.69–2.79) | 1.04 (0.79–1.38) | | 1.03 (0.82–1.46) | |
| | 45–55 | 104 | 2.19 (1.89–2.53) | 1 | | 1 | |
| | <45 | 46 | 2.11 (1.64–2.67) | 0.83 (0.63–1.09) | | 0.98 (0.68–1.43) | |
| Tobacco Smoking | Current smoker | 51 | 2.25 (1.82–2.76) | 1.12 (0.85–1.48) | 0.08 | 1.28 (0.92–1.77) | 0.04 |
| | Ex-smoker | 80 | 2.37 (2.01–2.77) | **1.29 (1.02–1.62)** | | **1.45 (1.08–1.93)** | |
| | Never smoker | 78 | 1.85 (1.53–2.23) | 1 | | 1 | |
| Spirometry | Obstructive | 13 | 3.33 (2.42–4.47) | 1.24 (0.90 1.72) | 0.22 | **1.69 (1.16–2.46)** | **0.01** |
| | Restrictive | 18 | 1.71 (1.11–2.50) | 0.81 (0.52–1.22) | | 0.83 (0.54–1.29) | |
| | Normal | 65 | 2.07 (1.54–2.23) | 1 | | 1 | |
| Baseline MRC breathlessness score | 3–5 | 11 | 2.57 (1.63–3.86) | 1.35 (0.82–2.22) | **0.01** | | |
| | 2 | 55 | **2.64 (2.17–3.17)** | **1.37 (1.08–1.73)** | | | |
| | 1 | 135 | 1.88 (1.63–2.15) | 1 | | | |
| Recreational drug use, ever | Yes | 77 | 2.19 (1.92–2.50) | 1.22 (0.85–1.76) | 0.28 | | |
| | No | 127 | 2.03 (1.68–2.42) | 1 | | | |
| Recreational drug use, last 3 months | Yes | 51 | 2.11 (1.69–2.60) | 0.92 (0.51–1.37) | 0.67 | | |
| | No | 153 | 2.18 (1.93–2.46) | 1 | | | |
| Ethnicity | Black African / Caribbean | 23 | **1.61 (1.06–2.37)** | **0.64 (0.42–0.96)** | **0.01** | 1.07 (0.62–1.88) | 0.25 |
| | South Asian / other / mixed | 13 | 3.01 (1.50–5.39) | 1.39 (0.98–2.00) | | 1.40 (0.94–2.07) | |
| | White | 173 | 2.13 (1.90–2.38) | 1 | | 1 | |
| Educational attainment | Other | 20 | 2.27 (1.61–3.12) | 0.88 (0.56–1.39) | 0.91 | | |
| | Degree | 116 | 2.38 (2.07–2.73) | 0.94 (0.70–1.25) | | | |
| | A level | 32 | 1.89 (1.41–2.47) | 0.91 (0.63–1.31) | | | |
| | None/GCSE | 40 | 1.64 (1.23–2.13) | 1 | | | |
| Baseline SGRQ | >20 | 42 | 2.72 (2.17–3.38) | 1.39 (0.68–2.24) | 0.14 | | |
| | 10–20 | 28 | 2.52 (1.99–3.14) | 1.41 (0.69–2.89) | | | |
| | <10 | 96 | 1.84 (1.58–2.11) | 1 | | | |
| Current CD4 [a] | >500 | 94 | 1.98 (1.67–2.33) | 0.73 (0.37–1.42) | 0.27 | | |
| | 350–500 | 27 | 1.86 (1.35–2.51) | 0.58 (0.26–1.29) | | | |
| | <350 | 15 | 3.30 (2.23–4.72) | 1 | | | |
| Nadir CD4 [a] | 500+ | 17 | 1.86 (1.19–2.76) | 0.75 (0.38–1.48) | 0.28 | | |
| | 350–500 | 20 | 2.05 (1.40–2.90) | 0.84 (0.45–1.58) | | | |
| | 200–350 | 36 | 1.84 (1.37–2.41) | 0.93 (0.55–1.56) | | | |
| | <200 | 63 | 2.26 (1.78–2.72) | 1 | | | |

[a]: calculated for HIV positive participants only

[b]: Poisson regression

## Symptom severity during acute illness

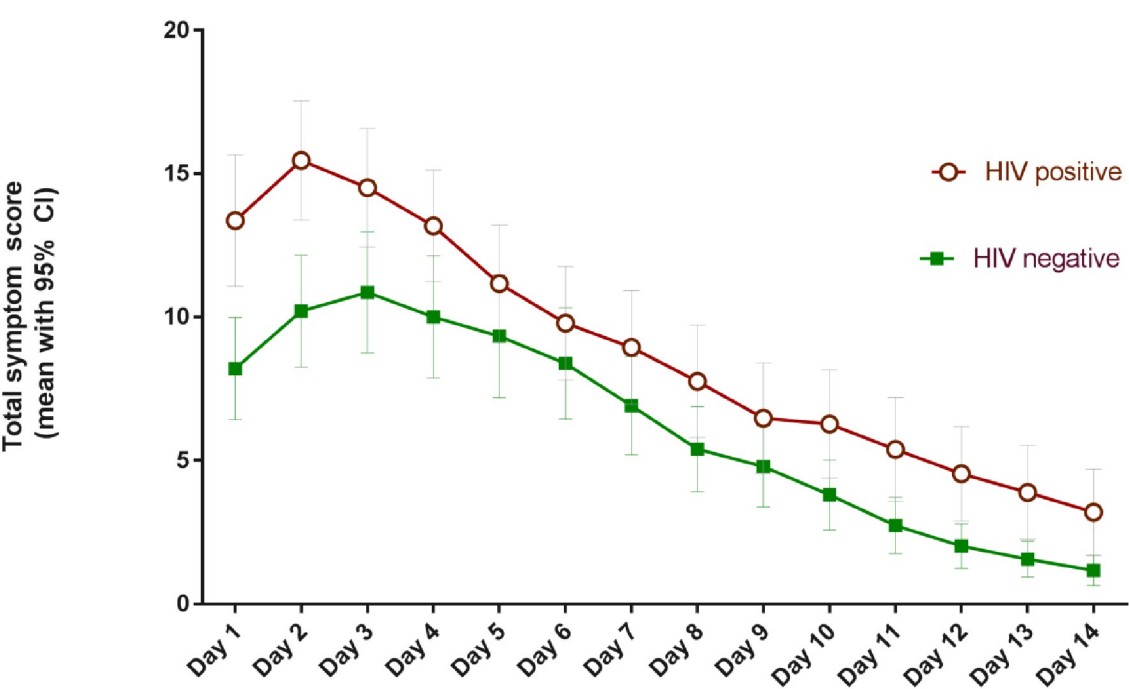

**Fig 1. Symptom severity during acute respiratory illness.**

### Treatment and healthcare utilisation

Using the diaries completed by participants, PLW-HIV were more likely to obtain advice from a healthcare professional during an acute respiratory illness (42% vs 14%, p = 0.003), and more likely to seek healthcare assessment 32% vs 9%, p = 0.001. There was no significant difference in the proportion of illnesses for which participants took non-prescription medications (59% vs 54%). PLW-HIV used antibiotics in a numerically greater proportion of illnesses (22% vs 11.5%), though this was not statistically significant (OR 2.11 (95% CI 0.75–5.94), p = 0.16).

### Isolation of viral and bacterial pathogens during acute respiratory illness

We collected 158 nasopharygeal swabs during an acute respiratory illness. Respiratory viruses were detected in 77 (48%) of these samples, with no significant difference in the likelihood of detection of respiratory viral pathogens between HIV positive and negative groups. A viral pathogen was identified in 52% of illnesses in PLW-HIV (49/94) compared to 45% of HIV negative participants (29/64), p = 0.36. Rhinovirus was the predominant virus identified in both groups, detected in 26% of samples from HIV positive participants and 28% of samples from HIV negative participants. Influenza was detected in 5 samples from PLWH (5%) and 1 sample from an HIV negative participant (2%), seasonal coronavirus was detected in 9 (10%) and 5 (8%) samples respectively, other respiratory viruses detected were enterovirus, Human metapneumovirus, parainfluenza, Respiratory Syncytial Virus, and adenovirus, all in one or two cases only. The relative proportion of different viral results are illustrated in Fig 2.

**Table 4. Relationship between participant baseline characteristics and symptom severity at time of reporting acute respiratory illness, log-scale linear regression analyses.**

| Characteristic | | Median (IQR) symptom score[#] | Univariable analysis | | Multivariable analysis | |
|---|---|---|---|---|---|---|
| | | | Fold-change in total symptom score[*] | P value[#] | Fold-change in total symptom score | P value[**] |
| HIV Status | HIV Positive | 14 (8–23) | 1.61 (1.28–2.02) | <**0.001** | **1.50 (1.14–1.97)** | **0.02** |
| | HIV Negative | 9 (5–14) | 1 | | 1 | |
| Gender | Female | 13 (7–22) | 1.08 (0.77–1.66) | 0.648 | 1.30 (0.92–1.83) | 0.15 |
| | Male | 11 (6–20) | 1 | | 1 | |
| Spirometry | Restrictive | 12 (6–23) | 1.00 (0.66–1.51) | 0.04 | 0.96 (0.69–1.35) | 0.56 |
| | Obstructive | 20 (10–23) | 1.51 (1.10–2.07) | | 1.16 (0.80–1.67) | |
| | Normal | 11 (6–19) | 1 | | 1 | |
| Recreational drugs (ever) | Yes | 12 (6–21) | 1.64 (0.81–1.36) | 0.71 | | |
| | No | 12 (6–20) | 1 | | | |
| Recreational drugs, past 3 months | Yes | 15 (8–21) | 1.34 (1.07–1.67) | **0.01** | 1.27 (0.95–1.72) | 0.27 |
| | No | 11 (6–20) | 1 | | 1 | |
| Ethnicity | Black African / Caribbean | 8 (4–21) | 0.83 (0.41–1.68) | 0.47 | | |
| | South Asian Other/Mixed | 13 (9–31) | 1.08 (0.48–2.43) | | | |
| | White / Caucasian | 12 (6–20) | 1 | | | |
| Qualifications / educational attainment | None | 6 (5–24) | 0.92 (0.69–1.22) | 0.07 | | |
| | GCSE | 8 (4–14) | 0.75 (0.55–1.02) | | | |
| | A level | 10 (7–17) | 0.78 (0.54–1.14) | | | |
| | Other | 13 (5–22) | 1.06 (0.58–1.53) | | | |
| | University /degree | 13 (6–21) | 1 | | | |
| Baseline St Georges Respiratory Questionnaire score | >20 | 14 (9–23) | 1.68 (1.30–2.18) | <**0.001** | 1.42 (1.11–1.83) | 0.05 |
| | 10–20 | 14 (8–21) | 1.48 (1.14–1.93) | | 1.43 (1.05–1.96) | |
| | <10 | 9 (4–18) | 1 | | 1 | |
| Baseline MRC Dyspnoea score | 3–5 | 14 (6–33) | 1.75 (1.03–2.98) | **0.004**[◊] | | |
| | 2 | 13 (7–21) | 1.22 (0.53–1.58) | | | |
| | 1 | 10 (5–18) | 1 | | | |
| Tobacco smoking | Current smoker | 11 (1–21) | 1.12 (0.83–1.51) | 0.22 | 0.91 (0.63–1.33) | 0.96 |
| | Ex-smoker | 11 (5–20) | 0.92(0.70–1.02) | | 1.06 (0.81–1.38) | |
| | Never smoker | 12 (6–19) | 1 | | 1 | |

[#] Univariable log-scale linear regression analysis

[*] univariable log-scale linear regression model

[**] multivariable log-scale linear regression including all factors with data in this column

[◊] MRC dyspnoea score not included in multivariable model as collinear with SGRQ score

We obtained sputum samples for bacterial culture during 70 illnesses: pathogens were identified in 10 (19%) of these samples from 9 different participants: *Haemophilus influenzae* (5 isolates from 4 participants), *Streptococcus pneumoniae* (3 isolates, each from different participants), *Moraxella catarrhalis* (2 from different participants). The fungus *Aspergillus fumigatus* was identified in one participant. All samples from which bacterial or fungal organisms were isolated were from HIV positive participants.

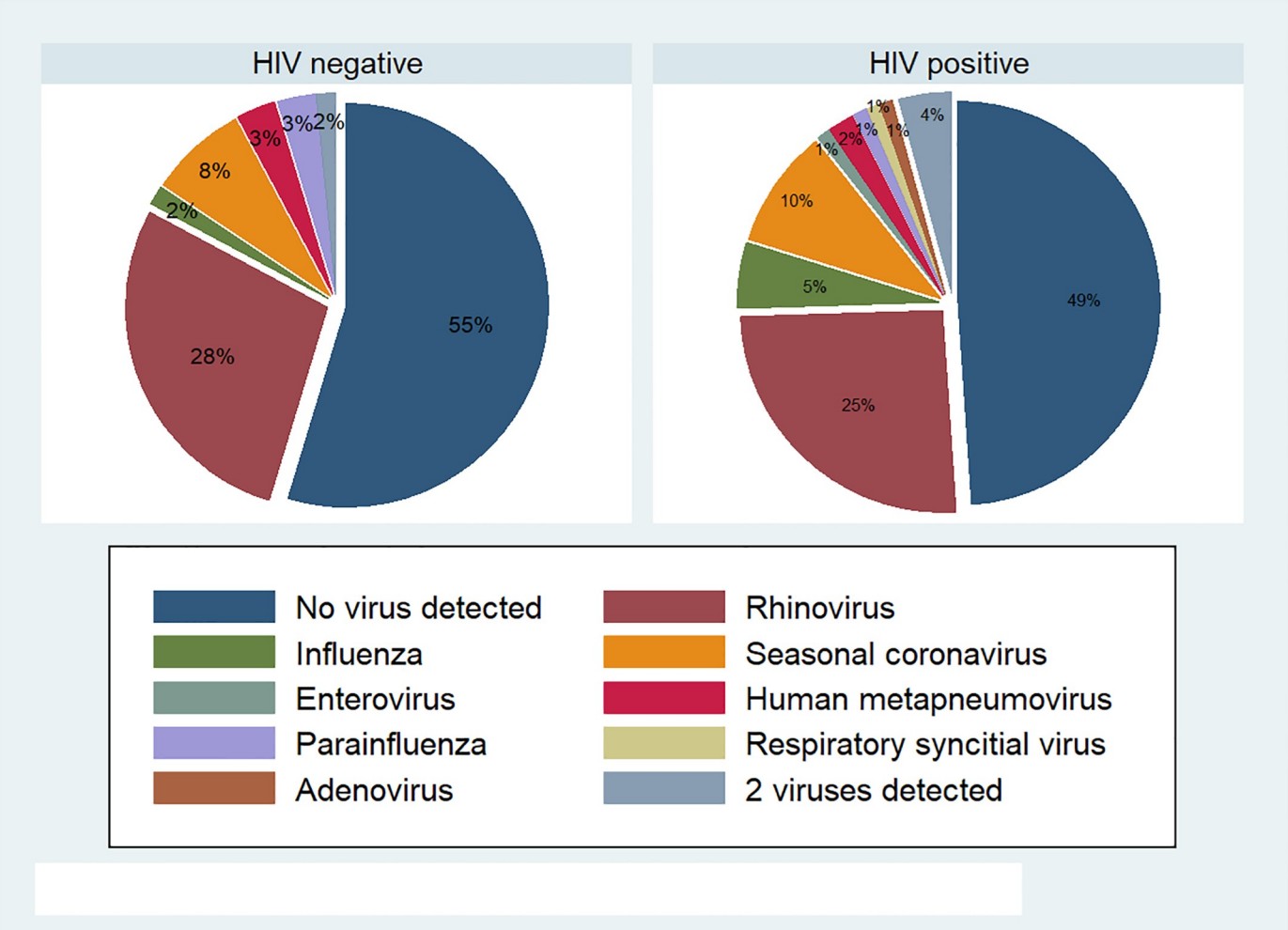

**Fig 2. Respiratory viruses detected during acute respiratory illness.**

## Sensitivity analyses

We conducted several pre-specified sensitivity analyses to ensure that our findings were robust using different analytical methods. These included: a) excluding all participants with less than 80% response rate; b) defining the offset value for the regression analyses as the number of weeks between the first and last response to the weekly messages recorded (rather than the total number of responses) and c) defining the outcome as the proportion of weeks of follow up in which a new respiratory illness was reported (thus giving a continuous rather than count variable) and performing linear regression analyses. The findings of the main analysis were consistent in all these sensitivity analyses. We also repeated the main analyses of frequency of events using a negative binomial model and findings were not significantly different to the Poisson model presented.

## Discussion

In this prospective cohort study, we found no evidence of a significant difference in the frequency of acute respiratory illness between HIV positive and negative individuals in a setting

with a high uptake of antiretroviral therapy. Our study was powered to detect a difference of 50% or more in the proportion of HIV positive participants experiencing an acute respiratory illness during follow up, however the tight confidence intervals around the incidence rate ratio between the two groups make it unlikely that there is more than a small difference in the frequency of these events.

PLW-HIV did report more severe symptoms during acute illnesses, which was associated with a greater likelihood of seeking healthcare advice and a higher (although not statistically significant) proportion of illnesses treated with antibiotics. This effect persisted after adjustment for important potential confounding factors (such as differences in tobacco smoking), but other differences between the HIV positive and negative groups (such as differences in immunisation frequency) could have influenced this outcome.

As noted elsewhere,[12] PLW-HIV in this population were more likely to report chronic respiratory symptoms at baseline. A greater burden of baseline respiratory symptoms was independently associated with the severity of acute respiratory illnesses during follow-up. However, this did not explain all of the difference between HIV positive and negative participants, as PLW-HIV's higher symptom scores during acute respiratory illnesses persisted after adjustment for baseline symptoms.

Only a small proportion of the PLW-HIV evaluated had airflow obstruction, in contrast to some other published studies, it should be noted that we only undertook pre-bronchodilator spirometry and this proportion might have been lower still in post-bronchodilator measurements. We did find a greater proportion of the PLW-HIV with lower than predicted vital capacity, a finding which is consistent with some other studies reporting the influence of HIV status on prevalence of lung function impairment, for instance that of Ronit *et al* evaluating a population in Denmark reported a mean absolute difference in FVC of 395mls,[25] a very similar difference to that found in this study.

There are several possible explanations for the greater burden of respiratory symptoms during acute respiratory illness. These include PLW-HIV having impairments in lung function that are not measured by spirometry (such as low diffusing capacity,[26] or small airways disease [27]). An awareness of the potential for more severe illness might result in an increased perception or concern about physical symptoms in PLW-HIV—which itself may be driven by the higher prevalence of anxiety or depression reported in this population.[28, 29] Finally, there is evidence to suggest that despite immune reconstitution with ART, PLW-HIV may still have greater immune activation than HIV negative individuals [30, 31]–and thus the heightened symptom burden during an acute respiratory illness might reflect a disordered immune response when unwell.

Although, this is the first prospective study of its kind, the greater use of antibiotics during acute illness is consistent with reports of a greater frequency of COPD exacerbations among PLW-HIV with COPD when these are defined by a requirement for treatment.[14, 15] Our study was not powered to address differences in antibiotic usage, though the results suggest that PLW-HIV may be more likely to receive antibiotic treatment for acute respiratory illnesses. This has potential implications for antimicrobial resistance.

We believe our results are relevant to both clinical care and health policy: the greater severity of acute respiratory illness among PLW-HIV suggests that (in addition to the provision of ART) there is a need to implement interventions that can improve the respiratory health. Our data suggest that it is possible to identify individuals at greater risk of more frequent or severe acute respiratory illness who could be the focus of interventions that can reduce their chance of respiratory illness—including smoking cessation, immunisation against influenza and *Streptococcus pneumoniae*—and improve the diagnosis and treatment of underlying respiratory

conditions. A simple risk assessment for respiratory illness could in future be part of routine HIV care assessments in people on antiretroviral therapy.

Our data also highlight the need to understand why PLW-HIV report more severe respiratory symptoms, both at baseline and during acute respiratory illness compared to matched controls. Identifying whether such differences in patient-reported outcomes reflect underlying pathological responses is important; and mechanistic studies which assess immune responses during acute illness might identify causes for the reported differences.

We aimed to document the frequency of all acute respiratory illness, and our study was not powered to measure less common but more severe events such as bacterial pneumonia. Epidemiological data suggest that pneumonia continues to be more frequent among PLW-HIV, and remains an important cause of mortality.[11, 17] Whether HIV positive people with good response to antiretroviral therapy continue to be at greater risk of severe bacterial pneumonia requires further attention.

Our study was based on self-report of acute respiratory illness, and utilised participant-determined measures to record respiratory symptoms, rather than an objective illness severity assessment. A limitation of our methodology was the lack of a standardised Patient Reported Outcome Measure for assessing acute respiratory illness severity, so a questionnaire was specifically created for this study. An illness definition and symptom questions were chosen with the aim of recording all acute respiratory illnesses, but this did not differentiate between upper and lower respiratory tract illnesses. We used intensive active surveillance to identify acute respiratory illnesses, including weekly contact with study participants, thus minimising recall and reporting biases. However, although the response rate to the weekly study contacts was very good (85%), written daily diaries during acute illnesses were only available in 46% of episodes. This may have introduced a degree of bias into our data, though it is reassuring to note that a greater severity of illness was reported from both the online and written diary respondents.

As with any study evaluating an HIV cohort, our findings are not necessarily generalizable to settings where access to HIV care is poor, or to individuals who do not maintain linkage to healthcare. Also, the pattern of utilisation noted by us may be less applicable to regions with different healthcare systems.

## Conclusions

In an adult population with a high uptake of antiretroviral therapy, there is no difference in the frequency of acute respiratory illness between HIV positive and negative individuals. When these do occur, people living with HIV report more severe and longer-lasting illnesses and are more likely to seek healthcare advice. This has implications for healthcare resource utilisation and the health-related quality of life of people living with HIV.

## Supporting information

**S1 Data.**
(PDF)

**S2 Data.**
(PDF)

**S3 Data.**
(PDF)

**S4 Data.**
(PDF)

**S5 Data.**
(PDF)

## Author Contributions

**Conceptualization:** James Brown, Elisha Pickett, Memory Sachikonye, Lucy Brooks, David M. Lowe, Sara Madge, Mike Youle, Margaret Johnson, John R. Hurst, Timothy D. McHugh, Ibrahim Abubakar, Marc Lipman.

**Data curation:** James Brown.

**Formal analysis:** James Brown, Colette Smith, Tabitha Mahungu, David M. Lowe, Timothy D. McHugh, Marc Lipman.

**Funding acquisition:** James Brown, Marc Lipman.

**Investigation:** James Brown, Elisha Pickett, Memory Sachikonye, Lucy Brooks, Sara Madge, Mike Youle, John R. Hurst, Timothy D. McHugh, Marc Lipman.

**Methodology:** James Brown, Colette Smith, Memory Sachikonye, Tabitha Mahungu, David M. Lowe, Margaret Johnson, John R. Hurst, Timothy D. McHugh, Ibrahim Abubakar.

**Project administration:** James Brown, Elisha Pickett, Lucy Brooks.

**Supervision:** David M. Lowe, Margaret Johnson, Ibrahim Abubakar, Marc Lipman.

**Writing – original draft:** James Brown.

**Writing – review & editing:** James Brown, Marc Lipman.

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
