## [Decision Letter · Decision Letter 0]

15 Jan 2020

PONE-D-19-35060

The effect of HIV status on the frequency and severity of acute respiratory illness

PLOS ONE

Dear Dr Brown,

Thank you for submitting your manuscript to PLOS ONE. After careful consideration, we feel that it has merit but does not fully meet PLOS ONE’s publication criteria as it currently stands. Therefore, we invite you to submit a revised version of the manuscript that addresses the points raised during the review process.

We would appreciate receiving your revised manuscript by Feb 29 2020 11:59PM. To enhance the reproducibility of your results, we recommend that if applicable you deposit your laboratory protocols in protocols.io, where a protocol can be assigned its own identifier (DOI) such that it can be cited independently in the future. For instructions see: http://journals.plos.org/plosone/s/submission-guidelines#loc-laboratory-protocols

We look forward to receiving your revised manuscript.

Kind regards,

Rachel M Presti

Academic Editor

PLOS ONE

Journal Requirements:

2. Please include additional information regarding the questionnaires used in the study and ensure that you have provided sufficient details that others could replicate the analyses. For instance, if you developed a questionnaire as part of this study and it is not under a copyright more restrictive than CC-BY, please include a copy, in both the original language and English, as Supporting Information. If the questionnaires have been published previously, please provide relevant references.

3. Please provide a sample size and power calculation in the Methods, or discuss the reasons for not performing one before study initiation.

4. Thank you for stating the following beneath the Acknowledgments Section of your manuscript:

'Funding: This study was supported by grants from the National Institute for Health Research (DRF-

2015-08-210) and British HIV association. The funders had no role in the collection, analysis, or

interpretation of data, in the writing of the report or in the decision to submit the paper for

publication.'

'JB received grant funding to complete this study from the National Institute for Health research; DRF-2015-08-210.

The funders had no role in study design, data collection and analysis, decision to publish, or preparation of the manuscript'

6. Please include a separate caption for each figure in your manuscript.

7. Your ethics statement must appear in the Methods section of your manuscript. If your ethics statement is written in any section besides the Methods, please move it to the Methods section and delete it from any other section. Please also ensure that your ethics statement is included in your manuscript, as the ethics section of your online submission will not be published alongside your manuscript.

Reviewers' comments:

Reviewer's Responses to Questions

**Comments to the Author**

1. Is the manuscript technically sound, and do the data support the conclusions?

Reviewer #1: Partly

Reviewer #2: Yes

2. Has the statistical analysis been performed appropriately and rigorously? 

Reviewer #1: No

Reviewer #2: Yes

3. Have the authors made all data underlying the findings in their manuscript fully available?

Reviewer #1: Yes

Reviewer #2: Yes

4. Is the manuscript presented in an intelligible fashion and written in standard English?

Reviewer #1: Yes

Reviewer #2: Yes

5. Review Comments to the Author

Reviewer #1: This prospective study by Brown Colleagues exams the incidence and severity of acute respiratory illnesses in an HIV-infected population on ART and compares it to non HIV-infected subjects. They report that the incidence of acute infections is similar in the two groups, but that the severity is worse in the HIV-infected subjects leading to greater seeking of care from the health care system by this group.

In general this is an easy to read, well written manuscript. The size of the two populations is reasonable. Follow up of the subjects over a year is very good for these types of studies. The main concerns relate to how the data was analyzed as described below.

Major Comments:

1. It appears there were two attempts at objective assessment of upper respiratory tract severity. The first was a “web-based symptom questionnaire” graded on a “0-6 scale, including questions on daily activities and treatment.” The second was a subject “diary”, which also appears to have been scored somehow given the data in the results section was also given as “points”. It is unclear how these two measurement tools are different. It would be nice to show these grading systems in supplemental materials to better assess the quality of the data. Furthermore, it needs to be clear in the methods that the first tool is actually web-based. In the methods section it read like the investigators were doing the question asking, which obviously introduces bias if the interviewer knows the subject status. It was not apparent that the first tool was web-based till the first sentence under the “Severity and Duration of Illness” section in the results.

2. There are a lot of differences between the two groups (Table 1). This is important because their adjustment for potential confounding effects was based on “a priori” factors known to be associated with acute respiratory illnesses rather than actual differences between the two groups. Thus, while baseline smoking, baseline respiratory symptoms, and baseline PFTs were appropriately considered, it is not clear that all the significant differences between the groups were taken into account in their univariate and multivariate analysis. Differences that were adjusted for that might be significant include:

a. UK born versus other – relates to different potential prior exposures and cultural responses to illness which might impact assessment of disease severity.

b. Immunizations – much higher in the HIV positive group, which may have lessened the incidence in the HIV positive group.

c. Significantly higher incidence of inhaler use, especially corticosteroids, in the HIV group. These by themselves can affect the incidence and severity of respiratory illnesses.

3. One of the author’s conclusions is that it is “possible to identify individuals at greater risk of more frequent or severe acute respiratory illness (using the degree of airflow obstruction, history of smoking or recreational drug use, and the presence of chronic respiratory symptoms). Does this hold for both HIV-infected subjects and uninfected subjects, or just the HIV population alone? From the tables, it looks like these risk factors were lumped together regardless of HIV status.

Minor Comments:

1. At the end of the Statistical analysis it states that further details are provided in supplemental information. I do not see any supplemental information. Based on major comment 2 above, this is important.

2. While this is in general very well written, for some reason there are multiple grammatical errors in the 4th paragraph of the discussion.

3. Which of the two severity measures is being used in Figure 1?

Reviewer #2: This is a very robust epidemiologic study looking at respiratory symptoms in a cohort of people living with HIV (PLwHIV) and relatively matched controls over a several year period of fairly robust regular communication. Respiratory illness was intentionally loosely defined to capture events that may result in communication with a healthcare professional or treatment. St. Georges Respiratory Questionaire (SGRQ) and the MRC dyspnea scale data was collected at baseline and during any acute illness. In addition an unweighted symptom score was collected daily during acute illness recovery. The PLwHIV were well controlled with all on ART and the majority with suppressed viral load. An attempt was made to identify the common viruses that were representative of the acute illnesses but this may be biased as more severe illness may result in less willingness to return for a swab. The control group is mostly age and gender matched with Ethnicity, sexuality, immunizations and respiratory medication use being different. The subjects had pre-bronchodilator spirometry and both restrictive and obstructive defects were more common in the PLwHIV. Associated with this both the SGRQ and MRC favored more symptoms in PLwHIV. The main outcome of the study, rate of acute illness was the same in both groups with most of the significant findings being related to the subjective measures of respiratory symptom severity. Consistent with would be predicted obstruction, prior smoking, and baseline dyspnea were all associated with greater illness frequency. Although the symptoms scores were higher and PLwHIV contacted medical providers more frequently use of non-prescription medications was equivalent. Overall 158 np swabs were evaluated and there was no difference in the rate of positive swabs between groups or in type of virus recovered. Overall the data is robust and discussed with appropriate notation of the limitations.

1. The incidence of “restrictive” lung disease seems quite large in comparison to prior studies. This would be more appropriately called this PRISm (preserved ratio impaired spirometry) given the individuals only had pre-bronchodilator spirometry. This likely would also be consistent with a predilection for increased symptoms and likely an increased risk of future incident COPD. I do not think this has previously been described. This may be worth adding to the discussion given the findings of the Rotterdam Study would be consistent. From this standpoint it may be important to note who (trained respiratory therapist, research coordinator, nurse) performed the spirometry as an important caveat would be poor effort can give this sort of finding as well but would not explain the association only with PLwHIV.

2. Likey discussion should include the fact that the spirometry was all prebronchodilator and likely over represents the population as having COPD.

3. Inclusion of more information on the symptom score that was utilized at illness initiation and daily diaries would be beneficial. The presumption is that this symptom score was de novo invented and likely has not been confirmed to be of value as an unscaled entity in this disease process. Although this does not lessen the value of this information, knowing that the subscales of the SGRQ were all significantly more in PLwHIV suggests knowing more information about what makes up the Figure 1 results would help.

4. Figure 2 is not particularly well presented. The coloring is based on the percentage of each virus but likely would be improved if both “pie” graphs utilized the same color for same viruses. It is not clear to me if there is any difference in any measured virus as I suspect as the only notation in the text is that Rhinovirus is the most common in both groups and many viral swabs were negative. For instance it seems that influenza vaccination is more common in PLwHIV but yet isolating influenza (probably not significant) was more common in PLwHIV. Given this is probably the most novel piece of data in this study it would be nice if it were slightly better presented.

5. oropharyngeal swabs. Page 4 and 10 states oropharyngeal swabs were utilized all other references suggest these were nasopharyngeal swabs. Likely this is a typo to correct. Suspect they all should be nasopharyngeal given it was a PCR assay.

6. A comment is made that tabulation of “febrile” illness rate is possible and likely it would be worthy to comment how this related to the subsets of acute illnesses and viral recovery from nasopharyngeal swabs.

7. The one factor that may be overlooked is the comfort with which PLwHIV may feel in interacting with health care professionals that may differ from the control population. It is possible that there is a bias towards reporting symptoms in this population given ongoing drug side effects and discussion of them not only establish rapport with health care professionals but may also have resulted in beneficial interactions. There seems to be proof of this in that the use of over the counter medications was equivalent despite the fact that more contact with providers and actual prescribed medications was higher in PLwHIV. Whether this is conditioned by health care providers or as stated a manifestation of mild COPD and PRISm would be nearly impossible to separate as finding a control population with a similar degree of health care exposure would be impossible.

6. PLOS authors have the option to publish the peer review history of their article (what does this mean?). If published, this will include your full peer review and any attached files.

Reviewer #1: No

Reviewer #2: No

---

## [Author Response · Author response to Decision Letter 0]

9 Mar 2020

We thank the reviewers for their detailed evaluation of our manuscript and constructive suggestions for improvement. We have amended the text in light of these comments, and provided responses to each point below.

2. Please include additional information regarding the questionnaires used in the study and ensure that you have provided sufficient details that others could replicate the analyses. For instance, if you developed a questionnaire as part of this study and it is not under a copyright more restrictive than CC-BY, please include a copy, in both the original language and English, as Supporting Information. If the questionnaires have been published previously, please provide relevant references.

Response: We have included both the baseline and follow-up questionnaires as supplementary material

 3. Please provide a sample size and power calculation in the Methods, or discuss the reasons for not performing one before study initiation.

Response: We have included a short section in the methods explaining our sample size calculation

4. Thank you for stating the following beneath the Acknowledgments Section of your manuscript:

'Funding: This study was supported by grants from the National Institute for Health Research (DRF-

 2015-08-210) and British HIV association. The funders had no role in the collection, analysis, or

 interpretation of data, in the writing of the report or in the decision to submit the paper for

 publication.'

Response: we have completed the funding statement in the online submission form. 

Response: We have uploaded anonymised data to the University College London data repository (https://rdr.ucl.ac.uk/) with DOI 10.5522/04/11950284, this will be made available on acceptance of the manuscript for publication.

6. Please include a separate caption for each figure in your manuscript.

Response: we have ensured that all figures have a separate caption. 

 7. Your ethics statement must appear in the Methods section of your manuscript. If your ethics statement is written in any section besides the Methods, please move it to the Methods section and delete it from any other section. Please also ensure that your ethics statement is included in your manuscript, as the ethics section of your online submission will not be published alongside your manuscript.

Response: Our ethics statement is included in the methods section of the manuscript

Comments to the Author

 Major Comments:

1. It appears there were two attempts at objective assessment of upper respiratory tract severity. The first was a “web-based symptom questionnaire” graded on a “0-6 scale, including questions on daily activities and treatment.” The second was a subject “diary”, which also appears to have been scored somehow given the data in the results section was also given as “points”. It is unclear how these two measurement tools are different. It would be nice to show these grading systems in supplemental materials to better assess the quality of the data. Furthermore, it needs to be clear in the methods that the first tool is actually web-based. In the methods section it read like the investigators were doing the question asking, which obviously introduces bias if the interviewer knows the subject status. It was not apparent that the first tool was web-based till the first sentence under the “Severity and Duration of Illness” section in the results.

Response: Thank-you for this suggestion. We have made it clearer in the methods section that follow-up data on the frequency of acute respiratory illness, symptoms during acute respiratory illness and their severity were collected by means of a web-based questionnaire (for most participants) and also written diaries (only completed during illnesses). We have also included these questionnaires as supplementary material. 

 2. There are a lot of differences between the two groups (Table 1). This is important because their adjustment for potential confounding effects was based on “a priori” factors known to be associated with acute respiratory illnesses rather than actual differences between the two groups. Thus, while baseline smoking, baseline respiratory symptoms, and baseline PFTs were appropriately considered, it is not clear that all the significant differences between the groups were taken into account in their univariate and multivariate analysis. Differences that were adjusted for that might be significant include:

 a. UK born versus other – relates to different potential prior exposures and cultural responses to illness which might impact assessment of disease severity.

 b. Immunizations – much higher in the HIV positive group, which may have lessened the incidence in the HIV positive group.

 c. Significantly higher incidence of inhaler use, especially corticosteroids, in the HIV group. These by themselves can affect the incidence and severity of respiratory illnesses.

Response: We agree with the reviewer that prior exposures and cultural differences, immunisation frequencies and differences in treatments such as inhaled corticosteroids could act as confounders in the analysis. However, we chose not to take the approach of defining adjustment models based on post hoc differences between groups defined by p values (an approach criticised in recent methodological reviews e.g. Lederer et al Annals ATS 2018) and our relatively small cohort lacks the statistical power to allow adjustment for all potential confounding factors. We have added a comment in the discussion that residual confounding (rather than a direct HIV related effect) could influence this finding. 

 3. One of the author’s conclusions is that it is “possible to identify individuals at greater risk of more frequent or severe acute respiratory illness (using the degree of airflow obstruction, history of smoking or recreational drug use, and the presence of chronic respiratory symptoms). Does this hold for both HIV-infected subjects and uninfected subjects, or just the HIV population alone? From the tables, it looks like these risk factors were lumped together regardless of HIV status.

Response: this comment in the discussion is based on analyses of the cohort as a whole, rather than just the HIV positive participants. On reflection, as implied by the reviewer, this probably goes beyond what can be supported by the data, as using the features identified in the analysis to guide care would in any case require further validation. We have therefore taken out this specific conjecture from the discussion (although continue to make the general point that individuals a high risk of respiratory illness can be identified and prioritised for interventions. 

Minor Comments:

1. At the end of the Statistical analysis it states that further details are provided in supplemental information. I do not see any supplemental information. Based on major comment 2 above, this is important.

Response: apologies for this error – there in no supplementary file for the statistical methods, although we have attached copies of our research questionnaires as supplementary files. 

2. While this is in general very well written, for some reason there are multiple grammatical errors in the 4th paragraph of the discussion.

Response: we thank the reviewer for this general positive comment and agree that this specific paragraph perhaps became rather convoluted during the drafting process, and have tried to make the prose easier to follow. 

3. Which of the two severity measures is being used in Figure 1?

Response: this is based on the scores calculated from the written diaries completed by participants during respiratory illnesses. we have made this clearer in the manuscript. 

Reviewer #2: This is a very robust epidemiologic study looking at respiratory symptoms in a cohort of people living with HIV (PLwHIV) and relatively matched controls over a several year period of fairly robust regular communication. Respiratory illness was intentionally loosely defined to capture events that may result in communication with a healthcare professional or treatment. St. Georges Respiratory Questionaire (SGRQ) and the MRC dyspnea scale data was collected at baseline and during any acute illness. In addition an unweighted symptom score was collected daily during acute illness recovery. The PLwHIV were well controlled with all on ART and the majority with suppressed viral load. An attempt was made to identify the common viruses that were representative of the acute illnesses but this may be biased as more severe illness may result in less willingness to return for a swab. The control group is mostly age and gender matched with Ethnicity, sexuality, immunizations and respiratory medication use being different. The subjects had pre-bronchodilator spirometry and both restrictive and obstructive defects were more common in the PLwHIV. Associated with this both the SGRQ and MRC favored more symptoms in PLwHIV. The main outcome of the study, rate of acute illness was the same in both groups with most of the significant findings being related to the subjective measures of respiratory symptom severity. Consistent with would be predicted obstruction, prior smoking, and baseline dyspnea were all associated with greater illness frequency. Although the symptoms scores were higher and PLwHIV contacted medical providers more frequently use of non-prescription medications was equivalent. Overall 158 np swabs were evaluated and there was no difference in the rate of positive swabs between groups or in type of virus recovered. Overall the data is robust and discussed with appropriate notation of the limitations.

 1. The incidence of “restrictive” lung disease seems quite large in comparison to prior studies. This would be more appropriately called this PRISm (preserved ratio impaired spirometry) given the individuals only had pre-bronchodilator spirometry. This likely would also be consistent with a predilection for increased symptoms and likely an increased risk of future incident COPD. I do not think this has previously been described. This may be worth adding to the discussion given the findings of the Rotterdam Study would be consistent. From this standpoint it may be important to note who (trained respiratory therapist, research coordinator, nurse) performed the spirometry as an important caveat would be poor effort can give this sort of finding as well but would not explain the association only with PLwHIV.

Response: spirometry was undertaken by a trained respiratory specialist, or clinical research practitioner with specific training in undertaking spirometry. Results were quality assured according to ATS guidelines. The finding of reduced lung volumes, without a difference in prevalence of airflow obstruction, described as restrictive spirometric pattern in our manuscript is consistent with findings of other studies reporting spirometric results in people living with HIV, for instance that of Ronit et al, Thorax 2019. We prefer the term restrictive spirometric pattern, which is more widely recognised than preserved ratio impaired spirometry, and avoids additional use of abbreviations in the text. We have added a paragraph to the discussion discussing the spirometry results in more detail 

 2. Likey discussion should include the fact that the spirometry was all prebronchodilator and likely over represents the population as having COPD.

Response: we have added this to the discussion

 3. Inclusion of more information on the symptom score that was utilized at illness initiation and daily diaries would be beneficial. The presumption is that this symptom score was de novo invented and likely has not been confirmed to be of value as an unscaled entity in this disease process. Although this does not lessen the value of this information, knowing that the subscales of the SGRQ were all significantly more in PLwHIV suggests knowing more information about what makes up the Figure 1 results would help.

Response: We have included the symptom scores and questionnaire as supplementary information. As the reviewer suggests, these questionnaires were devised de novo as we felt that existing scores did not meet the requirements of this study and we did not undertake a separate process of validation. 

 4. Figure 2 is not particularly well presented. The coloring is based on the percentage of each virus but likely would be improved if both “pie” graphs utilized the same color for same viruses. It is not clear to me if there is any difference in any measured virus as I suspect as the only notation in the text is that Rhinovirus is the most common in both groups and many viral swabs were negative. For instance it seems that influenza vaccination is more common in PLwHIV but yet isolating influenza (probably not significant) was more common in PLwHIV. Given this is probably the most novel piece of data in this study it would be nice if it were slightly better presented.

Response: Thank-you, we agree that this could be better presented. We have revised the text to include the important results in the text and replaced Figure 2 with a simple pie chart for illustrative purposes. 

 5. oropharyngeal swabs. Page 4 and 10 states oropharyngeal swabs were utilized all other references suggest these were nasopharyngeal swabs. Likely this is a typo to correct. Suspect they all should be nasopharyngeal given it was a PCR assay.

Response: as suggested, we collected nasopharyngeal swabs for respiratory virus detection, this has been corrected in the manuscript.

 6. A comment is made that tabulation of “febrile” illness rate is possible and likely it would be worthy to comment how this related to the subsets of acute illnesses and viral recovery from nasopharyngeal swabs.

Response: although we agree with the reviewer that further understanding of the relationship between illness characteristics such as fever and the identification of viral pathogens would be interesting, this is unfortunately beyond the data available – we only have data on self-reported fever, and have viral swab results for only 125 illnesses. Within this group there is no significant relationship between reported fever and detection of a virus (in the 33% of illnesses in which a fever of at least “moderate” severity was reported, a virus was detected 50% of the time; in the 67% of cases in which fever was not significant a virus was detected 41% of the time). The data was not collected with this analysis in mind, and hence we don’ t think this can be taken further. 

 7. The one factor that may be overlooked is the comfort with which PLwHIV may feel in interacting with health care professionals that may differ from the control population. It is possible that there is a bias towards reporting symptoms in this population given ongoing drug side effects and discussion of them not only establish rapport with health care professionals but may also have resulted in beneficial interactions. There seems to be proof of this in that the use of over the counter medications was equivalent despite the fact that more contact with providers and actual prescribed medications was higher in PLwHIV. Whether this is conditioned by health care providers or as stated a manifestation of mild COPD and PRISm would be nearly impossible to separate as finding a control population with a similar degree of health care exposure would be impossible.

Response: we agree that differences in ease of access to healthcare, and greater concern about symptoms in the PLW-HIV could account for differences in health-seeking behaviour, and have added more explicit discussion of this to the text (although we agree that this something that it would be difficult to fully evaluate)

---

## [Decision Letter · Decision Letter 1]

27 Apr 2020

The effect of HIV status on the frequency and severity of acute respiratory illness

PONE-D-19-35060R1

Dear Dr. Brown,

We are pleased to inform you that your manuscript has been judged scientifically suitable for publication and will be formally accepted for publication once it complies with all outstanding technical requirements.

With kind regards,

Eduard J Beck, PhD, FAFPHM, FFPH, FRCP

Academic Editor

PLOS ONE

Additional Editor Comments (optional):

Reviewers' comments:

Reviewer's Responses to Questions

**Comments to the Author**

1. If the authors have adequately addressed your comments raised in a previous round of review and you feel that this manuscript is now acceptable for publication, you may indicate that here to bypass the “Comments to the Author” section, enter your conflict of interest statement in the “Confidential to Editor” section, and submit your "Accept" recommendation.

Reviewer #1: All comments have been addressed

Reviewer #2: All comments have been addressed

2. Is the manuscript technically sound, and do the data support the conclusions?

Reviewer #1: Yes

Reviewer #2: Yes

3. Has the statistical analysis been performed appropriately and rigorously? 

Reviewer #1: Yes

Reviewer #2: Yes

4. Have the authors made all data underlying the findings in their manuscript fully available?

Reviewer #1: Yes

Reviewer #2: Yes

5. Is the manuscript presented in an intelligible fashion and written in standard English?

Reviewer #1: Yes

Reviewer #2: Yes

6. Review Comments to the Author

Reviewer #1: (No Response)

Reviewer #2: All issues have been addressed. No further changes are recommended. Figures are significantly improved

7. PLOS authors have the option to publish the peer review history of their article (what does this mean?). If published, this will include your full peer review and any attached files.

Reviewer #1: No

Reviewer #2: No

---

## [Editor Report · Acceptance letter]

20 May 2020

PONE-D-19-35060R1 

The effect of HIV status on the frequency and severity of acute respiratory illness 

Dear Dr. Brown:

I am pleased to inform you that your manuscript has been deemed suitable for publication in PLOS ONE. Congratulations! Your manuscript is now with our production department. 

With kind regards,

on behalf of

Dr. Eduard J Beck 

Academic Editor

PLOS ONE